# Pulmonary Inhalation of Biotherapeutics: A Systematic Approach to Understanding the Effects of Atomisation Gas Flow Rate on Particle Physiochemical Properties and Retained Bioactivity

**DOI:** 10.3390/pharmaceutics16081020

**Published:** 2024-08-01

**Authors:** Laura Foley, Ahmad Ziaee, Gavin Walker, Emmet O’Reilly

**Affiliations:** SSPC the SFI Research Centre for Pharmaceuticals, Department of Chemical Sciences, Bernal Institute, University of Limerick, V94 T9PX Limerick, Ireland; laura.foley@ul.ie (L.F.); ahmad.m.ziaee@ul.ie (A.Z.); gavin.walker@ul.ie (G.W.)

**Keywords:** biotherapeutics, spray-drying, atomisation gas flow rate, enzymatic activity, particle engineering, pulmonary delivery

## Abstract

The identification of spray-drying processing parameters capable of producing particles suitable for pulmonary inhalation with retained bioactivity underpins the development of inhalable biotherapeutics. Effective delivery of biopharmaceuticals via pulmonary delivery routes such as dry powder inhalation (DPI) requires developing techniques that engineer particles to well-defined target profiles while simultaneously minimising protein denaturation. This study examines the simultaneous effects of atomisation gas flow rate on particle properties and retained bioactivity for the model biopharmaceutical lysozyme. The results show that optimising the interplay between atomisation gas flow rate and excipient concentration enables the production of free-flowing powder with retained bioactivity approaching 100%, moisture content below 4%, and D_50_ < 4 µm, at yields exceeding 50%. The developed methodologies inform the future design of protein-specific spray-drying parameters for inhalable biotherapeutics.

## 1. Introduction

In 2022, the biopharmaceutical market was valued at USD 389.3 billion. With a compound annual growth rate (CAGR) of 7%, it is expected to reach USD 720.8 billion by 2030 [1]. Biotherapeutics have the ability to enter therapy routes that have not yet been explored for the treatment of more complex disorders and diseases [2]. They exhibit significantly greater selectivity, specificity, and potency than small-molecule drugs along with fewer side effects. Since the introduction of biopharmaceutical drugs in 1982, to the first biologic approval in 1998, the US Food and Drug Administration (FDA) has approved a total of 340 biologic drugs to date, with many more undergoing the approval process [3,4,5]. As outlined on the FDA approval list, the majority of biotherapeutics on the market today exhibit a molecular weight of 12–25 kDa. Many recombinant growth factors have a molecular weight of approximately 18 kDa; examples include Retacrit, Abseamed, and Binocrit, used to treat anaemia, including anaemia associated with chronic renal failure. Various human growth hormones have a molecular weight of approximately 20 kDa such as Skytrofa and Accretropin, while Jetrea, used in the treatment of vitreomacular adhesion, has an approximate molecular weight of 25 kDa [6]. Biotherapeutics are commonly delivered in one of two ways, intravenous infusions, or subcutaneous injections, both of which require medical supervision or training. However, delivery routes such as intranasal and pulmonary delivery are emerging as preferred alternatives as less medical supervision is required [7]. As the portfolio of biotherapeutics on the market expands, so too does the requirement for innovative formulation and delivery approaches.

Pulmonary administration is a non-invasive route that aims to deliver drugs directly to the alveoli in the lungs for rapid adsorption. It is capable of delivering higher drug concentrations and offers greater bioavailability and improved therapeutic efficacy with minimal immunogenicity [8]. Dry powder inhalation (DPI) is a form of pulmonary delivery in which the administered drug is delivered as micronized particles, engineered with specific physical properties for pulmonary absorption. For a drug to reach the lower respiratory tract and pass through the alveoli and into the bloodstream, particles should have an aerodynamic diameter of 0.5–5.0 µm and a residual moisture content of less than 5% [9]. In pulmonary administration, the hepatic first-pass metabolism can be bypassed, ensuring high bioavailability since the lungs locally exhibit minimal metabolic activity. However, known drawbacks including poor dosage repeatability, local toxicity, and immunogenicity should be considered in the drug development phase.

Ziaee et al., previously outlined the challenges of biotherapeutic manufacturing [5]. This includes challenges associated with maintaining the chemical and physical stability of biotherapeutics throughout manufacturing, transportation, and storage. The long-term stability of biotherapeutics is significantly impacted by factors such as pH and residual moisture content. Dehydrating biotherapeutic formulations leads to improved long-term stability at increased temperatures and humidity due to lower molecular mobility and fewer intermolecular interactions [10]. Freeze drying has been the most frequently used technique for the solidification of biopharmaceuticals to date; however, it requires high energy consumption, long drying times, and exhibits limited particle engineering capabilities [11]. Spray-drying offers many advantages over freeze drying, including shorter process cycle times and the ability to process at atmospheric pressures. Excellent particle engineering capabilities and compatibility with continuous manufacturing principles further favour its use in industrial settings [12,13]. However, one of the major challenges associated with the spray-drying of biomolecules is the high temperature necessary for solvent evaporation. The removal of water/solvent from the spray droplet is the most critical step in the particle’s formation and subsequent morphology in the final formulation. Critical quality attributes such as moisture content, particle size, and morphology can be controlled and adjusted during the drying process. A moisture content of <5%, D_50_ of 1–5 µm, and yields exceeding 50% are general requirements for powders prepared for pulmonary delivery applications [14]. The particles’ aerodynamic diameter and morphology are critical design variables with respect to aerosol performance. Uniquely, spray-drying permits control of both particle size and morphology, thereby greatly facilitating particle engineering for dry powder inhalation applications. The morphology of the particles can be further explained by understanding the stages of particle drying within the spray dryer [13,15].

Protein-based biotherapeutics are larger in size, more complex, and more thermally labile compared to small-molecule pharmaceuticals. Although spray-drying is depicted as a gentle drying process, biotherapeutics are sensitive to the high liquid–gas interface tensions which they experience post-atomization. Atomisation gas flow rate has previously been identified as one of the parameters with the most influence on final formulation parameters, such as particle morphology, and is a significant contributor to protein denaturation [16,17,18,19,20]. Protein aggregation and denaturation are considerable challenges when spray-drying protein-based biopharmaceuticals. High residual moisture content and shear stress applied during the atomisation process are known contributors to protein aggregation and denaturation, resulting in reduced bioactivity, heightened immunogenicity, and reduced therapeutic efficacy [21,22].

Previous spray-drying work in the field of biotherapeutics has been focused on the prevention of protein degradation through the optimisation of gas flow rate or the addition of excipients. The use of spray-drying for particle engineering has predominantly been investigated for small-molecule pharmaceuticals. The processing of biotherapeutics suitable for delivery via dry powder inhalation is dependent upon the development of spray-drying strategies that engineer particles to specific criteria while simultaneously maintaining molecule bioactivity by preventing protein degradation. The ability to control particle size and morphology is critical in processing biotherapeutics for delivery via dry powder inhalation. For efficient and effective delivery, particles must be <5 µm in size to pass through the membranes in the deep lung and enter the bloodstream.

This study investigates the simultaneous effect of atomisation gas flow rate on particle physical properties and bioactivity with a view to producing biotherapeutics for dry powder inhalation applications. Lysozyme, with a molecular weight of 14 kDa, is a representative biotherapeutic currently on the market today (12–25 kDa) and, as such, was selected as the model protein for this study. Variation in the atomisation gas flow rate for a variety of protein/excipient formulations, and subsequent evaluation of retained bioactivity and physiochemical properties such as particle size/shape/morphology and moisture content, provides a greater understanding of the behaviour of protein-based biotherapeutics during the spray-drying process.

## 2. Materials & Methods

### 2.1. Materials

Lysozyme (14 kDa) was purchased from Huisun Pharma, Taizhou, China. D-(+)-Trehalose dihydrate was purchased from Merck Life Science Ltd., Dublin, Ireland. Ultra-pure type II Millipore water was obtained from a Milli-Q water purification system (Merck Millipore, Burlington, MA, USA) and used in all the studies.

### 2.2. Methodology

#### 2.2.1. Spray-Drying

The liquid feedstock solutions used for these tests were prepared using ultra-pure type II Millipore water and carried out in triplicate. A 5% *w*/*v* of lysozyme was used with the addition of varying trehalose (excipient) contents: 1:1, 1:2 and 2:1 *w*/*w*; the addition of the excipient caused an overall change in solid content depending on the amount added. Trehalose was the selected excipient as it can be processed at high temperatures and for its applications for pulmonary delivery. The Buchi B-290 mini spray dryer (BÜCHI Labortechnik AG, St. Gallen, Switzerland) was used in open loop mode. This was used in conjunction with a dehumidifier set at −4 °C. A two-fluid nozzle with a 0.7 mm nozzle tip diameter was used throughout the study.

Three atomisation gas flow rates were investigated: 473 L/h, 601 L/h, and 742 L/h, with a feed rate of 1.5 mL/min, and the aspirator set to 100% (35.0 m^3^/h or 35,000 L/h). All the operating parameters are outlined in Table 1. The process was first stabilised using the solvent, and, once stable, the feedstock was then pumped into the drying chamber. All the samples were collected after 20–25 min of spray-drying once the spray-drying process was complete.

#### 2.2.2. Enzymatic Activity

The spray-dried samples were collected and stored in the fridge (2–8 °C) and evaluated for enzymatic activity within 24 h of processing. Turbidity was monitored over time, noting the changes in suspension of *Micrococcus lysodeikticus* by UV-Vis spectrophotometry [23,24,25]. A phosphate buffer containing the bacterium with an absorbance of 0.6–0.7 at 450 nm was prepared as the substrate. The phosphate buffer was made fresh and chilled to 2–8 °C. The spray-dried samples were prepared using the phosphate buffer to a concentration of 400 units/mL. Blank samples of phosphate buffer were analysed and the decrease in A_450_ was tested over 5 min period. Activity testing was carried out by carefully pipetting 2.5 mL of the substrate suspension and 0.1 mL of the enzyme solution with the A_450_ over 5 min. The following equation was used to calculate the unit/mL of lysozyme:
Units/mL enzyme = ((∆A^450/minTest − ∆A^450/minBlank)(df))/((0.001)(0.1))(1)
where:

df = dilution factor,

0.001 = Change in absorbance (∆A_450_) as per the unit definition,

0.1 = Volume (in mL) of the enzyme solution.

#### 2.2.3. Moisture Content Analysis

The residual moisture content of all the spray-dried samples was measured using a Perkin-Elmer 4000 TGA (Thermal Gravimetric Analysis) analyser purchased from Perkin Elner Ireland Limited, Dublin, Ireland. The samples were heated from room temperature to 200 °C under a nitrogen atmosphere at a heating rate of 10 °C/min. The residual moisture content was calculated based on the weight loss of all the samples up to 150 °C.

#### 2.2.4. Scanning Electron Microscopy (SEM)

SEM analysis (Hitachi SU-70, Hitachi Ltd., Tokyo, Japan) was performed to evaluate the morphological configuration of the spray-dried lysozyme. The samples were coated with gold–palladium for 1 min at 20 mA and analysed using a beam intensity of 5 kV and a working distance of 10 mm.

#### 2.2.5. Density

Density was measured using a true density analyser, the micrometrics Accupyc II gas displacement pycnometry system (Micromeritics, Norcross, GA, USA) based on the standard USP 699 procedure. The true density of the spray-dried samples was measured at ambient temperatures using helium as the purging gas. A total of 10 cycles was used for each evaluation.

#### 2.2.6. Powder X-ray Diffraction (PXRD)

The determination of the powder structure was characterized by an Empyrean diffractometer (PANalyrical, Phillips, Cambridge, UK). The diffractometer was used at room temperature in reflection mode with Cu Kα radiation (λ = 1.5406 Å). The tube voltage and current were set to 45 kV and 40 mA, respectively. The step size used was 0.03° with a scan speed of 0.07°/s. All the samples were prepared by gently pressing the powders onto a silicon zero-background disc using a clean glass slide.

#### 2.2.7. Differential Scanning Calorimetry (DSC)

For the thermal analysis, a Netzech Polyma 214 (Netzsch, Selb, Germany) was used to analyse the thermal properties of the lysozyme and spray-dried samples. Approximately 5 mg of powder sample was weighed out and loaded into the hermetically sealed aluminium pans. The samples were heated at 10 °C/min under nitrogen flow.

#### 2.2.8. Particle Size Analysis (PSA)

The average particle size was analysed using the ImageJ 1.53k Java 1.8.0_172 Java-based image processing programme that utilises images collected from the SEM. Measuring 100–200 particles for each sample allowed the determination of the average particle size.

#### 2.2.9. Shear Cell Test

The flow functions (FF) of the powder samples were determined using the FT4 powder rheometer. The tests were carried out as described by Ziaee et al. [5] and were derived from methods originally developed by Wang et al., and Freeman [26,27]. A 9 kPa pre-consultation pressure method was used for all the tests.

#### 2.2.10. Design of 3-Factorial Contour Plot 

Contour plots were generated using JMP Pro 17 software from the data presented in Table 1.

## 3. Results and Discussion

### 3.1. Processing Parameters

Table 1 displays the spray-drying processing parameters used for each formulation. This table outlines the various input parameters, variables, and their corresponding outputs employed throughout the study. Three atomisation gas flow rates were examined, i.e., 473 L/h, 601 L/h, and 742 L/h. Other critical parameters affecting the integrity of the final powder formulations include outlet temperature and solution concentration. For samples 1–12, it was important to maintain the T_out_ at 50 °C; this was measured using the thermocouple embedded into the spray-drying. The T_in_ was set and adjusted to achieve the desired T_out_ value, as previously mentioned. T_out_ has been reported as a critical parameter for spray-drying biomolecules.

Samples 1–3 represent pure lysozyme samples at atomisation gas flow rates of 473 L/h, 601 L/h, and 742 L/h respectively. A significant loss of protein integrity was observed by increasing the atomisation gas flow rate for these three samples. While particle size is integral for pulmonary delivery applications, bioactivity retention and moisture content are critical output variables for biopharmaceutical formulations. To investigate the effect of the excipient on the loss of protein integrity, a disaccharide excipient was added at various ratios to the original formulation (samples 4–12). Trehalose has proven stabilising capabilities. Trehalose interacts with the various surface macromolecules allowing flexibility within the compound. This is one theory in how trehalose works at stabilising proteins preventing degradation and aggregation [28,29,30,31]. Finally, considering the learning from samples 1–12, the process was optimised in the final three spray-drying runs (samples 13–15).

### 3.2. Particle Size Distribution

Particle size distribution (PSD) was performed on all the spray-dried samples. The results outlined in Table 1 show a decrease in the D_50_ particle size values as the atomisation gas flow rates increase. The results show that the samples without the presence of an excipient (samples 1–3) produced powders with particle sizes ranging from ~4 µm to ~16 µm (D_10_–D_90_) with varying atomisation gas flow rates. Sample 3, processed at 743 L/h, produced the smallest particle size (D_10_) of ~4 µm; however, it exhibited a D_50_ value of ~9 µm (see Appendix A).

Figure 1 shows a selection of SEM images for samples 1–15. The SEM images for all the samples can be viewed in Appendix A. Figure 1A–C show the SEM images of spray-dried formulations 1–3. The surface of the particles seen in Figure 1A is noticeably less uniform and rougher, with more fragments and broken particles compared to those seen in Figure 1B,C. As the atomisation gas flow rate increases, the surface of the particles becomes smoother, and they appear more uniform. This can be attributed to the droplet size that is produced post-atomisation. Droplets that have a higher surface-to-volume ratio will undergo faster drying rates, and the Peclet number will increase. The Peclet number, which impacts particle formation, is described as the ratio of the diffusion coefficient of the solute and the evaporation rate [32]. Overall, this produces a smoother particle with fewer deformities [13]. The particles are hollow, permitting the respective wall thicknesses to be examined. As the atomisation flow rate and the temperature increase during the different spray-drying processes, the droplets at the higher atomisation rates dry quicker than those at the lower atomisation rates, forming a reduced particle wall thickness. A significantly thicker particle wall is formed when longer drying times are implemented, as the wet core dries slower. The samples exhibited a thickness of 5.7 µm (sample 1, spray-dried lysozyme at 473 L/h), 2.8 µm (sample 2, spray-dried lysozyme at 601 L/h), and 1.8 µm, respectively, (sample 3, spray-dried lysozyme at 742 L/h); SEM images highlighting particle wall thickness can be seen in Appendix A accompanying this paper. The results show that the wall thickness of the particles decreases with an increase in the atomisation gas flow rate.

Figure 1D–F, which contain a lysozyme to trehalose ratio of 2:1 *w*/*w*, have similar particle sizes to Figure 1A–C. These samples range from ~8 µm to ~11 µm, respectively, and can be seen in Table 1. However, the key difference between Figure 1A–F is the morphology. Figure 1D–F displays a dimpled indented morphology when compared to Figure 1A–C, in which the excipient is absent. Particles in Figure 1A–C are spherical in shape with a smooth morphology, whereas particles in Figure 1D–F are irregular in shape and dimpled in morphology [33]. Morphology is a key factor when determining the aerosolization of spray-dried samples. It has previously been reported that particles with smooth, flat surfaces have stronger adhesion forces between them, resulting in enhanced aggregation, which is undesirable for biotherapeutic formulations. In previous research, trehalose significantly increased the spray-dried powders’ dispersibility and had the highest inhalation performance when compared to other commonly used excipients like lactose and mannitol [30,34]. Various large biomolecules and proteins are often reported to exhibit hollow and dimpled particles post-spray-drying [35], which aligns with the observations in this study. Figure 1D–F shows a dimpled morphology for the samples containing trehalose as an excipient. Examining the SEM images, Figure 1D–F are also hollow. The flowability and cohesion of the optimised samples with the addition of trehalose were examined, and the results are reported in Section 3.7 below.

Understanding the interplay of processing parameters and particle size is critical in establishing the optimum processing conditions. Figure 2 shows a 3-factorial contour plot outlining particle size as a function of atomisation gas flow rate and trehalose concentration. The contour plot shows that while high atomisation gas flow rates are necessary for sub-5 µm particle sizes, careful consideration must be given to the excipient concentration. Trehalose concentrations of more than 50% of the total formulation increase particle sizes irrespective of atomisation gas flow rate, resulting in the particle size (D_50_) of the final formulation being quite large, ranging from ~5 to 26 µm (see Table 1: samples 4–12). The dark red region corresponds to the largest particle size, having a D_50_ average of ~26 µm and high particle agglomeration (Sample 12, lysozyme: trehalose 1:2 spray dried at 742 L/h), with the highest atomisation gas flow rate and trehalose to protein content. The dark green region has a particle size of <4 µm; both samples are spray-dried at the highest atomisation gas flow rate, however, the ratio of trehalose to protein has reduced.

### 3.3. Bioactivity Retention

It is well documented that moisture and temperature have profound effects on protein stability in the solid state. Changes to biotherapeutics due to physical and chemical factors (Sample 12, lysozyme: trehalose 1:2 spray-dried at 742 L/h) can affect the proteins’ folding and 3D structure, and subsequently the biotherapeutic efficacy, product quality, and safety. The key aim of this study is to identify the processing parameters to produce particles suitable for dry pulmonary inhalation applications while maintaining protein bioactivity. Therefore, trehalose was used as the stabilising excipient at various ratios outlined in Table 1. Trehalose, unlike other excipients, can be processed at high temperatures, thereby protecting the protein from being denatured or experiencing losses in bioactivity. More specifically, trehalose was chosen for its inhalation precedence and its high glass transition temperature of ~115 °C [36,37]. To date, most dry biological powders are maintained in the cold chain, where the product must be stored and transported sub-−20 °C [38,39]. Maintaining cold chain requirements is difficult and highly costly, especially for less developed regions and warmer climates. The introduction of excipients as part of the formulation enables the production of stable protein formulations in the dry form [40,41], a key requirement for dry powder inhalation.

The enzymatic activity of lysozyme and lysozyme–trehalose combinations were measured within 24 h of processing. Analysing samples within this period removes the possibility of protein refolding or reactivation and provides an independent assessment of the effects of the processing parameters. Figure 3A below compares the percentage of the bioactivity retention of samples 1–12 versus the unprocessed stock lysozyme. Pure spray-dried lysozyme in the solid form was increasingly deactivated as the atomisation gas flow rate increased. Sample 3 (orange bar, lysozyme spray-dried at 742 L/h) exhibited the lowest residual bioactivity of lysozyme after spray-drying. Alternatively, samples produced at the lowest atomisation gas flow rate (orange bar, lysozyme spray-dried at 473 L/h) showed the highest residual bioactivity (88%). These experiments demonstrated the direct effect of atomisation gas flow rate, i.e., shear force on the denaturation of lysozyme. Figure 3 also shows the significant impact of the addition of trehalose on the retention of enzymatic activity, at the highest atomisation gas flow rate. The addition of trehalose is essential in overcoming the negative effects on enzymatic activity induced by the shear stress of the atomisation gas flow rate. In Figure 3A, we can see that the 2:1 ratio of lysozyme: trehalose (purple bars) is the optimal ratio for retaining bioactivity, followed by the 1:1 formulation, and lastly the 1:2 formulation. When processing at an atomisation gas flow rate of 742 L/h, the addition of the trehalose excipient at this ratio resulted in a nine-fold increase in retained activity.

Figure 3B shows a 3-factorial contour plot combining all the results reported in Table 1. The dark red region represents the highest retention of bioactivity. There are two main regions where bioactivity retention is ~100%. Firstly, at the lowest atomisation gas flow, rate (473 L/h), 2:1 lysozyme: trehalose (sample 5), and at the highest atomisation gas flow, (742 L/h) 2:1 lysozyme: trehalose (sample 15). While both sets of processing parameters maintain enzymatic activity, it should be noted that they differ in terms of moisture content, as outlined in Table 1. Sample 5 (lysozyme: trehalose 2:1 spray dried at 473 L/h) has ~7% residual moisture present after spray-drying, while sample 15 (lysozyme: trehalose 2:1 spray dried at 105 °C and 742 L/h) has ~4%. Increased residual moisture content can lead to increased agglomeration occurring over time. The results herein confirm that of previous studies whereby the addition of excipients such as trehalose not only protects the biologic from shear stress and high temperatures but also prevents moisture-induced deterioration over time [42]. Additionally, increased residual moisture content has been reported to hinder the aerodynamic performance of the formulation [43].

As previously noted, the highest atomisation gas flow induces degradation of the protein structure as it is exposed to the air–liquid interface during atomisation. Proteins are amphiphilic, which can lead to their adsorption at the surface, exposing the hydrophobic core as it aligns at the interface. To reduce these effects, excipients can be used to directly compete for adsorption at the surface with the protein [44]. A study conducted by Wendorf et al. concluded that with an increase in sugar concentration, a decrease in protein adsorption is seen, which is in line with the results of this study [45]. Another study conducted by Webb et al. showed a loss of native protein structure could be avoided through the addition of a surfactant to reduce protein adsorption at the air–liquid interface during atomisation, thus improving the stability [42,46]. However, as the sugar content varies from 2:1 to 1:1 to 1:2 (lysozyme: trehalose), less protein is absorbed. Wendorf et al. concluded that with an increase in sugar concentration, a decrease in sugar adsorption is seen [45]. Therefore, less protein is being protected from the shear stresses and high temperatures experienced throughout the spray-drying process, leading to a decrease in retained bioactivity.

Trehalose is a highly stable disaccharide from both thermodynamic and kinetic perspectives. Burek et al. [47] suggest three key theories on why trehalose is a suitable means of protecting biological species in harsh processing environments. One theory concluded that due to the sugar and the protein not being mutually exclusive, its protective mechanism was adapted based on the abiotic stress factors it endures. The second is that trehalose can protect the protein during dehydration processes that would occur during the spray-drying process when water is removed in the drying process. Trehalose is composed of two glucose molecules bound by glycosidic bonds with a high degree of flexibility, which enable the sugar molecule to interact with the polar groups of the protein, in this case, lysozyme. The water-binding theory states that trehalose, rather than binding to the biomolecule directly, traps water close to the structure, keeping it in its natural state of hydration. The vitrification theory contends that trehalose develops into a non-hydroscopic glassy state with high stability and temperature resistance. The protein’s secondary, tertiary, and quaternary structure can be prevented from unfolding by the glassy matrix, which would otherwise result in a general loss of bioactivity [40,45,47,48,49].

### 3.4. Product Yield

Achieving acceptable product yields is a key component of any spray-drying process if industrial feasibility is to be realised, especially for high-cost biopharmaceuticals. As the retained moisture content and particle size increased, a reduction in drying efficiency was observed. This resulted in poor to moderate yields (<40%) being observed in samples 4–12. Despite the reduced yields observed, the bioactivity retention of samples 4–12 was dramatically improved with the addition of trehalose. Samples 13–15 demonstrate that by increasing the T_out_, a decrease in particle size and improvement in product yield was observed, without affecting retained bioactivity. Formulations retrieved from the final three samples that were spray dried at higher T_in_, produced samples with a particle size suitable for pulmonary delivery applications, along with high yields (~50–60%) and high bioactivity retention (~70%). A three-factorial contour plot for yield optimisation can be seen below. A higher yield is achieved when working in the green zones of the contour plot, right of Figure 4 below.

### 3.5. Differential Scanning Calorimetry (DSC)

DSC was employed to assess the thermal transitions of the spray-dried formulations. Measurement of variations in melting temperature (T_m_) can be used to determine the stability of a biotherapeutic–excipient formulation. Figure 5 below shows the DSC thermograms for each lysozyme–trehalose formulation (samples 4–15). The first endotherm exhibits a broad peak ranging from 50 to 110 °C, which can be attributed to the large internal heterogeneity and broad relaxation times of the protein samples [50]. The second endotherm (T_m_) is observed from 155 to 165 °C, depending on the concentration of trehalose present in the sample. The curves for samples 6, 9, and 12 contain the lowest amounts of trehalose (lysozyme: trehalose 1:2, spray-dried at 473, 601, and 742 L/h, respectively) and exhibit a Tm ~153 °C. Trehalose in isolation typically exhibits a Tm at 117.5 °C. The curves for samples 4, 7, and 10 contain a 1:1 ratio of lysozyme to trehalose sugar (spray-dried at 473, 601, and 742 L/h, respectively). The trehalose content results in a shift of the T_m_ endotherm curve to ~158 °C, closer to the endotherm for native lysozyme (202.6 °C). The samples containing a 2:1 ratio (protein: sugar) exhibit the highest T_m_ values of ~167 °C. The higher the T_m_, the more thermodynamically stable the protein is [51].

### 3.6. Residual Moisture

High levels of residual moisture negatively impact biochemical stability and the subsequent product shelf-life of biological therapeutics [52]. This predominantly occurs through increased rates of decomposition due to the conformational flexibility induced by the presence of water [53]. The residual moisture content of all the spray-dried samples was determined using TGA (see Table 1). The samples dried at a higher atomisation gas flow rate retained a lower percentage of residual moisture. As the atomisation gas flow rate decreases, the residual moisture in the sample increases. This is due to the formation of larger particles as the inlet temperature and gas flow rate decrease. This can also be attributed to slower drying kinetics resulting in incomplete drying cycles [5]. Enzymes and many protein-based biotherapeutics are hygroscopic, which is an important consideration during the manufacturing process. An efficient spray-drying process for biotherapeutics should produce a formulation with low moisture content and a high level of biological activity. Whereas high atomisation gas flow rates reduce moisture content, the increased shear force is a significant contributor to reduced biological activity and enzymatic degradation. In this study, processing at higher atomisation gas flow rates produced particles with the desired properties for pulmonary delivery applications, in addition to a lower tendency of agglomeration. Through optimisation of processing parameters and the addition of trehalose, lysozyme was sufficiently protected from thermal stresses, leading to an optimal formulation with a low moisture content and minimal loss of bioactivity. This can be seen in Table 1, samples 13–15. Sample 15 (represented by the blue dot in Figure 6, lysozyme: trehalose 2:1 spray dried at 105 °C and 742 L/h) produced a formulation with the lowest moisture content and highest yield, in addition to a particle size suitable for pulmonary delivery and a high retention of activity.

Figure 6A shows the relationship between atomisation gas flow rate and trehalose content, and (B) atomisation gas flow rate and inlet temperature in relation to the percentage of moisture content in the final formulation. The contour plots provide a unique way of understanding the interplay between individual processing parameters and assist in identifying alternative processing conditions. The dark green region highlights the optimum processing conditions for obtaining a formulation with the lowest moisture content (~3%) in addition to the other critical factors outlined above. The quality target profile for moisture content in a final formulation powder is <5%. Excessive processing temperature and shear force are key contributors to protein denaturation and should be minimised where possible. Figure 6B indicates that, with a reduced atomisation gas flow rate of ~601 L/h and a lower inlet temperature of 95 °C, a moisture content of ~5% can be obtained while maintaining a high yield and bioactivity retention (Sample 14, lysozyme: trehalose 2:1 spray-dried at 95 °C and 742 L/h). For effective processing of thermally labile materials such as biotherapeutics, understanding the interplay between processing parameters is critical in identifying the correct conditions for high bioactivity retention, small particle size, and high production yield.

### 3.7. FT4 Flow

During both pharmaceutical and biopharmaceutical manufacturing, the ease with which powders flow is a primary concern. This is especially pertinent in the processing of medicines for dry powder inhalation. Correlations between particle size/morphology and particle flowability are well established. Particles with excellent flowability are categorised as free-flowing in Table 2; these are favoured for dry powder inhalation applications for serval reasons. These include dosage accuracy, reduction in agglomeration potential, ease of patient use, and an increase in patient compliance. Powders with good flowability also have optimal aerosolization and manufacturing efficiency which are essential for deep lung penetration, and ease of handling and filling [54,55,56]. Employing continuous manufacturing techniques like spray-drying, both the particle morphology and size can be manipulated to target a specific final powder performance. Powder manufacturers can access the data needed to adopt a knowledge-led approach to powder property control and engineer the behaviours needed to increase process efficiency and produce a final formulation that is consistent and of high quality.

An FT4 flow instrument was used to identify the flow function and the cohesion value of the optimised powders (samples 13–15). This allowed us to investigate the flowability of the powders that were spray-dried at a 2:1 (protein: sugar at 742 L/h and 85–105 °C) ratio. Table 2 shows cohesion and flow function values for samples 13–15 and identifies the flow properties, with the samples being identified as either easy-flowing or free-flowing. For reference, samples with 4 < FF < 10 are considered to be easy-flowing samples, and samples with FF > 10 are considered free-flowing [57]. The high flowing capacity of these powders is due to their particle’s spherical, however dimpled, morphology. The spherical shape reduces contact time and assists in minimising cohesion [35,58]. When considered in tandem with density calculations, FT4 flow measurements provide important information on downstream powder processing parameters.

### 3.8. Density

The deposition of particles in the respiratory tract depends on both the physiochemical properties of the particle and the physiological conditions of the patient. The physiochemical parameters include moisture content, morphology, size, and bulk density while the physiological conditions refer to the patient’s breathing patterns and general lung health. Particles with a bulk density of ~1 gcm^−3^ are preferred for pulmonary drug delivery applications as they improve particle dispersibility and delivery efficiency [59]. Table 1 shows that the density stayed consistent throughout the process (1.35 g cm^−3^) with little variation. Samples 13–15 contained the experimentally determined optimum trehalose content. The density of trehalose is 1.58 g cm^−3^, and lysozyme, as received, is 1.36 g cm^−3^; therefore, the overall density of the formulation increased to 1.375 g cm^−3^ for each of these samples.

## 4. Conclusions

This study highlights the simultaneous effect of atomisation gas flow rate and inclusion of a sugar-based excipient on particle physiochemical properties and retained bioactivity of spray-dried lysozyme as a model biological molecule. Moreover, it identifies a systematic approach for engineering biotherapeutic particles with defined size, shape, and moisture content while simultaneously minimising protein degradation. If implemented for commercial biotherapeutics with a similar molecular weight range (12–25 kDa), the results herein can be used to inform the design of protein-specific spray-drying processing parameters, thereby significantly advancing the development of inhalable and non-invasive biotherapeutic drug delivery.

## Figures and Tables

**Figure 1 pharmaceutics-16-01020-f001:**
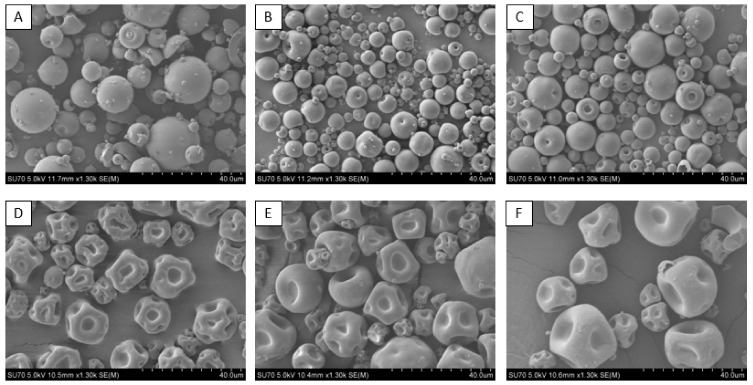
(**A**) SEM images of sample 1 spray-dried lysozyme at an atomisation gas flow rate of 473 L/h; (**B**) SEM images of sample 2 spray-dried lysozyme at an atomisation gas flow rate of 601 L/h; (**C**) SEM images of sample 3 spray-dried lysozyme at an atomisation gas flow rate of 742 L/h; (**D**) SEM images of sample 5 spray-dried lysozyme: trehalose 2:1 at an atomisation gas flow rate of 473 L/h; (**E**) SEM images of sample 8 spray-dried lysozyme: trehalose 2:1 at an atomisation gas flow rate of 601 L/h; and (**F**) SEM images of sample 11 spray-dried lysozyme: trehalose 2:1 at an atomisation gas flow rate of 742 L/h.

**Figure 2 pharmaceutics-16-01020-f002:**
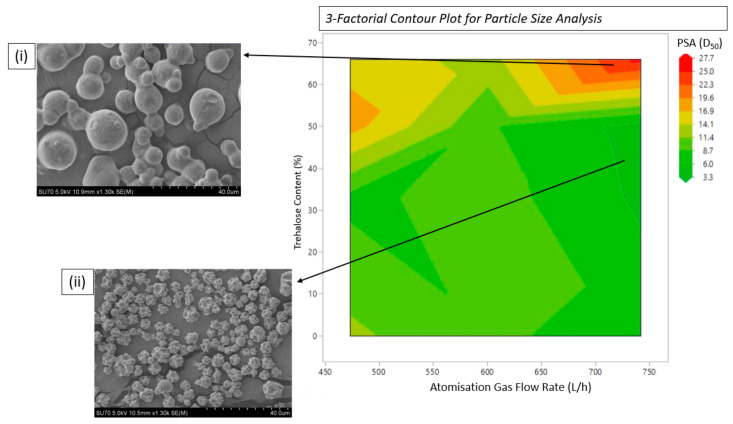
Effect of atomisation gas flow rate and trehalose content on particle size for all samples in Table 1. SEM images are captures of samples (**i**) 12 (lysozyme: trehalose 1:2 spray-dried at 742 L/h) and (**ii**) 15 (lysozyme:trehalose 2:1 spray-dried at 742 L/h).

**Figure 3 pharmaceutics-16-01020-f003:**
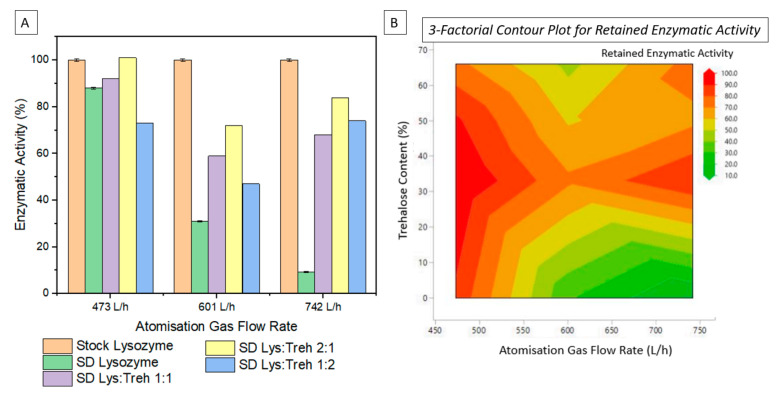
(**A**) Comparative graph of the retained activity of lysozyme before and after spray drying and with the addition of the excipient trehalose; (**B**) 3-factorial contour plot for retained enzymatic activity examining the effects of atomisation gas flow rate and trehalose content.

**Figure 4 pharmaceutics-16-01020-f004:**
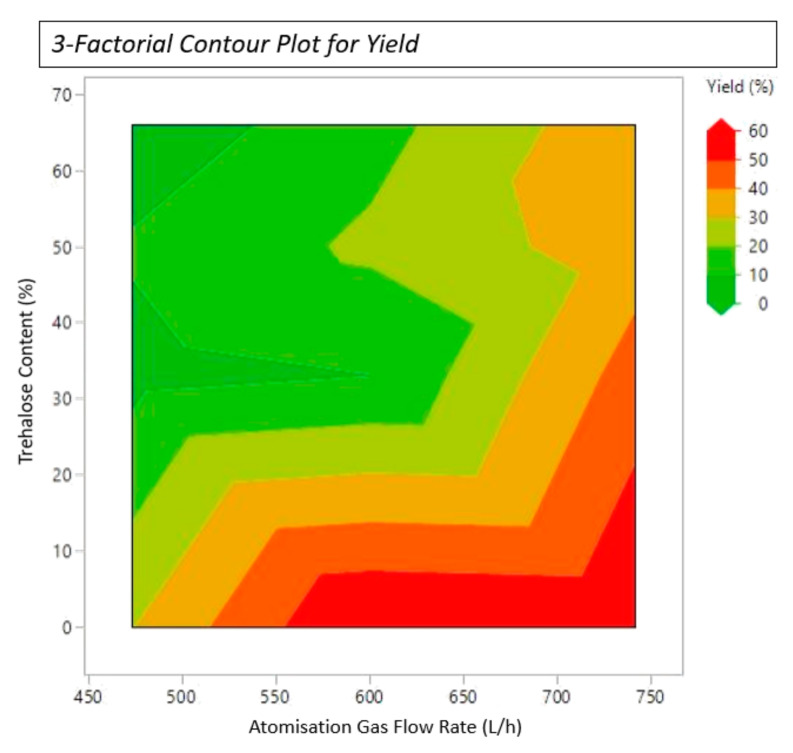
3-factorial contour plot for yield reflecting the effects of atomisation gas flow rate and trehalose content.

**Figure 5 pharmaceutics-16-01020-f005:**
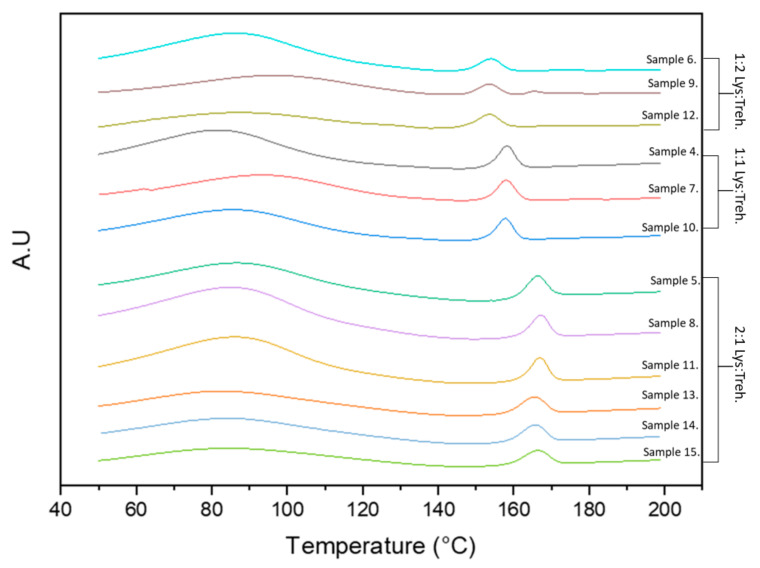
DSC thermograms of the spray-dried samples (4–15, in Table 1) of lysozyme with varying addition ratios of trehalose.

**Figure 6 pharmaceutics-16-01020-f006:**
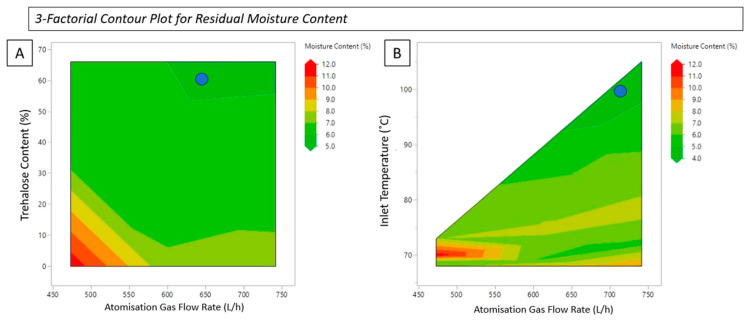
3-factorial contour plot for residual moisture content of the effect of (**A**) atomisation gas flow rate and trehalose content, and (**B**) atomisation gas flow rate and inlet temperature on the final residual moisture content of the spray-drying process. The blue dot in both graphs represents the optimum parameters to achieve the lowest moisture content.

**Table 1 pharmaceutics-16-01020-t001:** The initial process parameters and final formulation configurations of the spray-drying process.

ID	Inputs						Outputs					
	Atomisation Gas Flow Rate (L/h)	Inlet Temp (°C)	Outlet Temp (°C) **	Feed Rate (mL/min)	Solid Conc. (wt.%)	Ratio Lys:Treh (%)	Moisture Content (wt.%)	Yield (%)	Density (g/cm^3^) *	Particle Size		Enzymatic Activity (%)
										D_50_ (µm) *	Span	
1	473	70	50	1.5	5	-	11.7	29.5	1.36 [0.007]	12.2 [3.29]	0.7	88 [0.395]
2	601	74	50	1.5	5	-	7.2	61.7	1.37 [0.013]	9.6 [2.71]	0.9	31 [0.165]
3	742	78	50	1.5	5	-	7.4	60.2	1.35 [0.006]	8.9 [4.93]	1.3	9 [0.183]
4	473	73	50	1.5	10	1:1 (50:50)	6.9	11.1	-	18.4 [6.56]	1.2	92 [0.001]
5	473	68	50	1.5	7.5	2:1 (66:33)	6.7	6.6	-	8.2 [2.63]	0.9	101 [0.0002]
6	473	69	50	1.5	7.5	1:2 (33:66)	6.9	3.3	-	16.6 [7.06]	1.1	73 [0.0037]
7	601	72	50	1.5	10	1:1 (50:50)	6.3	21.9	-	9.2 [4.74]	1.3	59 [0.002]
8	601	72	50	1.5	7.5	2:1 (66:33)	6.2	9.9	-	10.2 [5.27]	1.1	72 [0.003]
9	601	69	50	1.5	7.5	1:2 (33:66)	6.0	16.2	-	13.8 [12.5]	1.3	47 [0.001]
10	742	72	50	1.5	10	1:1 (50:50)	6.6	35.3	-	5.2 [1.84]	0.9	68 [0.002]
11	742	68	50	1.5	7.5	2:1 (66:33)	9.2	8.4	-	10.9 [6.69]	1.5	84 [0.001]
12	742	72	50	1.5	7.5	1:2 (33:66)	4.8	37.0	-	26.1 [21.0]	1.7	74 [0.017]
13	742	85	62	1.5	7.5	2:1 (66:33)	6.3	56.4	1.38 [0.002]	4.9 [3.09]	1.5	99 [0.0004]
14	742	95	70	1.5	7.5	2:1 (66:33)	5.5	59.3	1.38 [0.003]	4.5 [3.18]	1.7	74 [0.002]
15	742	105	77	1.5	7.5	2:1 (66:33)	3.7	54.0	1.37 [0.005]	4.1 [1.91]	1.3	103 [0.002]

* [Standard deviation]. ** Set value.

**Table 2 pharmaceutics-16-01020-t002:** Flow functions and cohesion values derived via FT4 instrumentation.

ID	Cohesion KPa	Flow Function (FF)	Flow Properties
13	0.467	9.33	Easy-Flowing
14	0.454	9.52	Easy-Flowing
15	0.332	12.8	Free-Flowing

## Data Availability

The raw data supporting the conclusions of this article will be made available by the authors upon request.

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
