# Peer review of "Pulmonary Inhalation of Biotherapeutics: A Systematic Approach to Understanding the Effects of Atomisation Gas Flow Rate on Particle Physiochemical Properties and Retained Bioactivity"

_pharmaceutics, 2024, doi:10.3390/pharmaceutics16081020_

Round 1

Reviewer 1 Report

Comments and Suggestions for Authors

The article “Pulmonary inhalation of biotherapeutics; A systematic approach to understanding the effects of atomisation gas flow rate on particle physiochemical properties and retained bioactivity.” discusses spray drying of large molecule for preparation of dry powder inhaler. The article is well structured and written, discussing all the studies in detail. 

Minor suggestions/comments:

Line 226: The role of Peclet Number on particle morphology can be discussed in more detail for better context for readers. 

Did you perform aerosolization/inhalation testing on these powders to check whether the parameters such as mass median aerodynamic diameter (MMAD) and fine particle fraction (%FPF) are within the target value?

Reviewer 2 Report

Comments and Suggestions for Authors

This study examined the simultaneous effects of atomization gas flow rate on particle properties and retained bioactivity for the model biopharmaceutical lysozyme. Overall, the manuscript is well written with good data presentation, discussion and conclusion. However, following comments need to be considered to accept this manuscript.

Specific comments

Materials and Methods

2.2.1. Spray Drying

What was the excipient used in the formulation preparation? Please mention that and explain the reasons in this section.

Line 120, Table 1 doesn’t show 5% w/v solids for different ratio of lys:excipient formulation! In addition, the information about spray drying parameters mentioned in table 1 should be explained in the text. Why did you use different inlet temperatues? How did you decide to do 15 formulations? Is there any specific statistical design you used to determine number of formulations, if so explain in details?

Line 127, why did you collect the powder after 20-25 min of spray drying? What was the temperature and humidity during that waiting time?  

Results

Along with all the findings the data shows good flowability of the powder but to confirm the claim of the suitability of pulmonary inhalation delivery authors need to report the aerodynamic diameter and lung deposition study using impactor.

Reviewer 3 Report

Comments and Suggestions for Authors

Spray drying of biotherapeutics, as an alternative to lyophilization, has been an area of interest over the last few decades. Foley et al explored the feasibility of spray drying for lysozyme as a surrogate protein drug, and properties like size, bioactivity, and moisture content as response variables. By controlling atomization flow rate and excipient concentration, authors reported to have achieved desired particles with nearly 100% bioactivity, low moisture, and small size. While the research design, execution, results, rationale to the findings, and the overall presentation are great, it does not check/justify the novelty aspect. Several reports on spray drying with lysozyme as a surrogate are already available in literature. To name a few:

Elkordy et al (https://doi.org/10.1016/j.ijpharm.2004.02.027

Ji et al (https://doi.org/10.1016/j.ijpharm.2004.02.027

Liao et al (https://doi.org/10.1211/0022357021611

I would suggest the authors to highlight novelty and unique findings in this work. On these grounds, the submitted manuscript does not meet the journal standards in current form.

Reviewer 4 Report

Comments and Suggestions for Authors

The authors report spray drying experiments of feedstocks containing lysozyme and trehalose in several ratios and compare the characteristics of the powders obtained at 3 different values of the atomization gas flow rate. Most of the study (samples 4-12) was performed with T(outlet) kept at 50°C, an unfortunate choice since it resulted in "poor to moderate yields" (line 366). Some "optimized" samples (samples 13-15) were then prepared with higher T(outlet).

This study might still be of some interest for colleagues planning to carry out similar experiments. However, there are several points that must be addressed before the paper can be considered for publication.

1.      The authors need to provide information about the repeatability of their experiments. If it was not tested, or if it was tested only for some sets of parameters (such as the center point at 50% trehalose, 601 L/h), this needs to be clearly stated and commented on in the discussion

2.      The reporting of data in some tables and plots needs to be modified to ensure better clarity and consistency for the readers: see comments 3 to 8.

3.      In Table 1 and in the text, replace the Lys:Treh ratio by the percentage of trehalose, to ensure consistency with Figures 2, 3b, 4 and 6a

4.      This will allow to replace the current Sample IDs by more meaningful ones, e.g. "33%-473" instead of "5". The meaningful sample IDs can then be added in the SEM images of Figure 1 and 2 and in Figure 5 (not just in the legend!).

5.      Also, the sequence in Table 1 does not follow a logical order of trehalose content; for example, samples 4-5-6 (all spray-dried at 473 L/h) have 50wt%, 33 wt% and 66 wt% of trehalose, respectively, which does not allow for easy evaluation of the trends.  

6.      Similarly, the ordering of the samples in the histogram of Figure 3A does not follow the trehalose content.

7.      Once corrected, this histogram is a more appropriate type of graph than the contour plots created with JMP, because these contour plots create the misleading impression that they are built from more than 9 or 12 data points. At the very least, the position of the data points should be added on top of the colour maps.

8.      Line 320: samples 13-15 (at higher Toutlet) were added to Figure 3B. Is it also the case for the other graphs and, if so, how was this done? Feeding the data in a DoE would result in a skewed experimental space since T(outlet) was varied for only one trehalose percentage.   

Please also consider the typos or inconsistencies listed below:

9. The green-to-red scale needs to be reversed in Figure 3B (as was done in Figure 4), since a high value of the retained enzymatic activity is the desired result.

10. Line 218: D50 value is reported as 8 µm instead of 8.9 µm in Table 1

11.  Line 417: "trehalose formulations reduce the moisture content as they are highly hygroscopic" - to what does "they" refer in this sentence? Trehalose is not highly hygroscopic

12.  Line 437: sample 14 does not have moisture < 5%

13.  "Particle thickness" should be particle wall (or shell) thickness; the values should be rounded to a realistic number of significant figures (this also applies to moisture contents).

Round 2

Reviewer 3 Report

Comments and Suggestions for Authors

Accept.